# A Case of Diabetic Ischemic Ulcer with Toe Deformity Successfully Treated with Revascularization and Surgical Offloading

**DOI:** 10.3390/jcm14030646

**Published:** 2025-01-21

**Authors:** Kazuhito Nagasaki, Kyota Kikuchi, Masuomi Tomita, Katsuya Hisamichi, Yuko Izumi

**Affiliations:** 1Department of Vascular Surgery, Lower Limb Vascular and Wound Center, Shimokitazawa Hospital, Tokyo 155-0031, Japan; 2Department of Orthopedics, Lower Limb and Foot Diseases Comprehensive Center, Shimokitazawa Hospital, Tokyo 155-0031, Japan; kkikuchi@shimokitazawa-hp.or.jp; 3Department of Diabetology, Diabetes Center, Shimokitazawa Hospital, Tokyo 155-0031, Japan; mtomita@shimokitazawa-hp.or.jp; 4Department of Dermatology, Lower Limb and Foot Diseases Comprehensive Center, Shimokitazawa Hospital, Tokyo 155-0031, Japan; amazake7@gmail.com; 5Department of International Cooperation for Medical Education, International Research Center for Medical Education (IRCME), Graduate School of Medicine, The University of Tokyo, Tokyo 113-8655, Japan

**Keywords:** toe deformity, diabetic ischemic ulcer, revascularization, surgical offloading

## Abstract

**Background:** Diabetic ischemic ulcers with toe deformities are challenging to manage due to combined ischemia, infection, and mechanical stress. This case report highlights the successful treatment of a complex diabetic ischemic ulcer using a multidisciplinary approach that included revascularization and surgical offloading. **Case Presentation:** A 70-year-old male with type 2 diabetes mellitus presented with non-healing ulcers on the right third toe. The ulcers, located at the dorsal PIP joint and plantar MTP joint, were attributed to ischemia, infection, and progressive toe deformity. Angiography revealed significant arterial stenosis, which was treated with percutaneous transluminal angioplasty (PTA) to restore in-line flow and improve skin perfusion pressure. Surgical offloading included PIP resection arthroplasty and metatarsal shortening osteotomy. Postoperative management ensured complete ulcer healing, and no recurrence was observed during the three-year follow-up. **Discussion:** This case underscores the importance of combining revascularization to improve perfusion and surgical offloading to alleviate mechanical stress. Key factors for success included the restoration of in-line flow, achieving sufficient skin perfusion pressure, and reducing plantar pressure. Multidisciplinary collaboration among vascular surgeons, orthopedists, and wound care specialists played a critical role in achieving excellent long-term outcomes. **Conclusions:** Revascularization followed by surgical offloading provided effective treatment for a diabetic ischemic ulcer with toe deformity. This multidisciplinary approach demonstrates the necessity of individualized strategies to manage complex diabetic foot cases and prevent recurrence.

## 1. Introduction

Offloading therapy is pivotal in managing diabetic ischemic ulcers complicated by toe deformities. Although a total contact cast is the gold standard for offloading diabetic foot ulcers, it is contraindicated in ischemic limbs due to risks of tissue necrosis and ulcer deterioration from compression. Removable knee-high offloading devices are recommended for ischemic limbs as they allow effective pressure relief while enabling wound monitoring and infection control. However, surgical offloading becomes necessary when ulcers persist or recur despite adequate offloading. Surgical options include Achilles tendon lengthening, metatarsal head resection, and arthroplasty. The proper selection of surgical interventions should be based on a comprehensive assessment of the systemic condition, ischemic severity, deformity, ulcer progression, and infection status [1]. Diabetic foot ulcers (DFUs) complicated by ischemia and deformities present significant clinical challenges, requiring a combination of revascularization and offloading to achieve effective outcomes. However, the IWGDF 2023 guidelines lack specific criteria for blood flow evaluation and indications for surgical offloading [1]. In this case, in-line flow and SPP were prioritized as key parameters for assessing blood flow and guiding surgical offloading. This approach underscores the importance of individualized management strategies that address both ischemia and mechanical stress. The present report provides a detailed analysis of this strategy, contributing to the understanding of therapeutic interventions in complex DFU cases. Here, we present a case of a diabetic ischemic ulcer with toe deformity treated successfully with the combination of revascularization and surgical offloading.

## 2. Case Report

A 70-year-old male was diagnosed with type 2 diabetes mellitus 20 years ago. At the time of presentation, the HbA1c level was 6.8%, indicating moderate glycemic control. The diabetes was managed with oral antidiabetic medications, including metformin and sulfonylurea, along with a basal–bolus insulin regimen. The patient also had a history of dyslipidemia, managed with statins, diabetic neuropathy, peripheral arterial disease, and developed ischemic ulcers on the right first and second toes. He received PTA and the right first and second ray amputation at another hospital in June 2019. After the surgery, ankle–foot orthosis was dispensed, and routine callus management was used. However, poor orthotic adherence, worsening ischemia, and the progression of toe deformities led to new ulcer formations on the dorsum of the right third toe and the plantar surface of the third metatarsal head. In November 2020, he was referred to our hospital for further management.

### 2.1. Clinical Findings

The ulcers measured 18 × 15 mm on the dorsal PIP joint and 20 × 15 mm on the plantar aspect of the third metatarsal head, which extended to the fascia with surrounding erythema. A radiographic examination showed no osteolysis in the proximal phalanx or metatarsal bones. Pedal pulses were faint, and the ankle–brachial index (ABI) was 0.70. SPP was 35 mmHg dorsally and 30 mmHg plantarly. Based on the above, according to the WIfI classification, the ulcer at the PIP joint and on the plantar aspect of the third metatarsal head can be classified as W1I2fI2 (Stage 4) and W2I2fI2 (Stage 4), respectively. Plantar pressure mapping revealed high-pressure areas at the third toe’s MTP joint (6.61 kPa/cm^2^) (Figure 1a).

### 2.2. Management Plan

Revascularization was prioritized to improve blood flow to the ulcerated area, as insufficient perfusion was identified as the primary barrier to wound healing and the success of surgical offloading. Post procedure, the patient’s SPP values normalized within 48 h, confirming the success of revascularization. Antibiotic therapy was continued to manage infection, and surgical offloading was planned once the signs of infection subsided. One week after revascularization, infection control was achieved. We ultimately decided to perform surgical offloading to address mechanical stress and promote long-term ulcer healing. These interventions were selected based on a multidisciplinary discussion involving vascular surgeons, orthopedists, and wound care specialists. The patient’s vascular status, foot biomechanics, and overall clinical condition were carefully evaluated to ensure the optimal timing and sequencing of treatments.

### 2.3. Revascularization

Angiography revealed 75% stenosis in the anterior tibial artery, 90% stenosis in the dorsal foot artery, and occlusion in the posterior tibial artery’s distal branches (Figure 2a,b). PTA was performed using PTA balloons of varying sizes (Figure 2c), achieving restored in-line flow to the third toe’s plantar arteries (Figure 2d,e).

### 2.4. Surgical Offloading

PIP resection arthroplasty and metatarsal shortening osteotomy were performed to reduce pressure at the ulcer sites. The PIP joint was accessed dorsally, infected tissues were excised, and the head of the proximal phalanx was resected (Figure 3 (upper left)). A 45°oblique osteotomy was performed at the distal third of the metatarsal (Figure 3 (upper right)). The third metatarsal head was shortened by sliding it proximally, and internal fixation with a K-wire ensured stabilization (Figure 3 (lower figure)).

### 2.5. Follow-Up

After surgery, a standardized dressing protocol was implemented, using non-adherent hydrocolloid dressings changed every 3 to 4 days under sterile conditions. Antibiotic therapy was administered for 10 days to prevent infection. The postoperative course was uneventful, with no signs of wound infection. Sutures were removed at two weeks post surgery, and the K-wire was removed at three weeks post surgery, following the confirmation of satisfactory wound healing and stabilization. The plantar pressure at the MTP joint decreased to 3.54 kPa/cm^2^ (Figure 1b). During the treatment phase, the patient wore therapeutic sandals to offload pressure from the surgical sites. After complete wound healing, custom-made orthopedic shoes with pressure-redistributing insoles were provided to prevent ulcer recurrence. These shoes were used consistently during the maintenance phase and adjusted annually based on clinical evaluations. At a three-year follow-up, the patient remained ulcer-free with no recurrence of deformity or new ulcerations. The ABI remained within normal limits, with no significant radiographic findings during the follow-up. This indicated the sustained effectiveness of revascularization in maintaining adequate perfusion. This comprehensive postoperative management strategy, including dressing care, therapeutic footwear, and routine perfusion monitoring via the ABI, played a pivotal role in achieving the successful long-term outcomes observed in this case.

## 3. Discussion

Claw toe deformities are common in diabetic patients and are further complicated compared to non-diabetic individuals because of unique biomechanical and histological changes associated with the disease due to neuropathy, prolonged hyperglycemia, and intrinsic muscle atrophy [2]. These deformities exacerbate plantar pressure, significantly increasing the risk of ulceration. Elevated plantar and joint pressures, combined with external forces from walking and footwear, lead to pain and callus formation. The mechanism involves hyperextension at the MTP joint, causing the metatarsal head to protrude into the plantar surface. Concurrently, the anterior migration of the metatarsal fat pads decreases the cushioning effect, further elevating plantar pressure. Additionally, in diabetic neuropathy, plantar skin hardens, and soft tissue between the skin and bone becomes thinner, exacerbating the increases in plantar pressure [2,3]. Fryberg et al. emphasized that plantar pressure exceeding 6 kgf/cm^2^ significantly increases the risk of ulcer formation, particularly in patients with neuropathy and foot deformities [4]. Although specific offloading thresholds lack clear evidence, reducing excessive plantar loading is critical to prevent complications such as callus-induced ulcers, osteomyelitis, and even limb amputation [3,5,6]. In this case, the plantar pressure at the ulcer site was high (6.61 kPa/cm^2^) before surgery, indicating a high risk of ulcer formation due to walking load. The patient was difficult to manage even with conservative offloading; therefore, surgical offloading was deemed necessary.

Key considerations include ensuring adequate blood flow and controlling infection, which are critical before any surgical offloading is performed in ischemic limbs. Revascularization plays a pivotal role in improving wound healing potential, particularly through the restoration of in-line flow and achieving SPP values predictive of successful outcomes (≥50 mmHg) [7]. In-line flow, as described in the “Woundosome” concept, emphasizes the direct and continuous arterial blood flow to the wound site via major arterial pathways or effective collateral routes, which are essential for tissue repair and regeneration [8]. Elevated plantar pressure at ulcer sites (≥6.61 kPa/cm^2^) necessitated surgical offloading to redistribute mechanical stress. Post surgery, the reduction in plantar pressure to 3.54 kPa/cm^2^ aligned with the goals of offloading, significantly contributing to ulcer prevention and improved long-term outcomes [2]. Additionally, surgical offloading was chosen over toe amputation for this patient to preserve the biomechanical integrity of the foot and minimize the risk of adjacent toe deformities or ulceration, which are common complications following toe amputation. Quebedeaux, T.L. et al. have shown that toe amputation, particularly of the great toe, disrupts the foot’s biomechanics, leading to a shift in plantar pressure distribution toward the lateral toes [6]. This shift increases the likelihood of adjacent toe deformities such as claw toe and hammertoe, as well as the development of new ulcers due to abnormal mechanical stress on previously unburdened areas [5,6].

Multidisciplinary collaboration was pivotal in achieving the excellent outcomes in this case. Vascular surgeons ensured optimal revascularization, while orthopedic specialists addressed structural deformities through surgical offloading. Infection control measures, combined with tailored offloading strategies, minimized postoperative complications and facilitated effective wound healing. Importantly, surgical offloading must address the risk of postoperative infection, particularly in ischemic limbs where impaired perfusion increases susceptibility to wound complications. Adequate blood flow was ensured through revascularization prior to surgery, and the use of implants was avoided to minimize infection risks. Techniques such as K-wire fixation, which reduce reliance on implants, were selected as they align with infection prevention strategies. These measures were instrumental in achieving infection-free healing and optimizing the patient’s mobility and quality of life [2].

## 4. Conclusions

Revascularization followed by surgical offloading provided excellent outcomes in a diabetic ischemic ulcer with toe deformity. This case underscores the need for individualized, multidisciplinary management strategies to optimize patient outcomes in complex diabetic foot cases. Restoring in-line flow, achieving sufficient SPP, and addressing elevated plantar pressure through surgical offloading were crucial for long-term success. Effective collaboration between vascular surgeons, orthopedists, and wound care specialists played a central role in achieving infection-free healing, ulcer prevention, and improved patient mobility. These findings emphasize the importance of a comprehensive, team-based approach in treating challenging diabetic foot conditions.

## Figures and Tables

**Figure 1 jcm-14-00646-f001:**
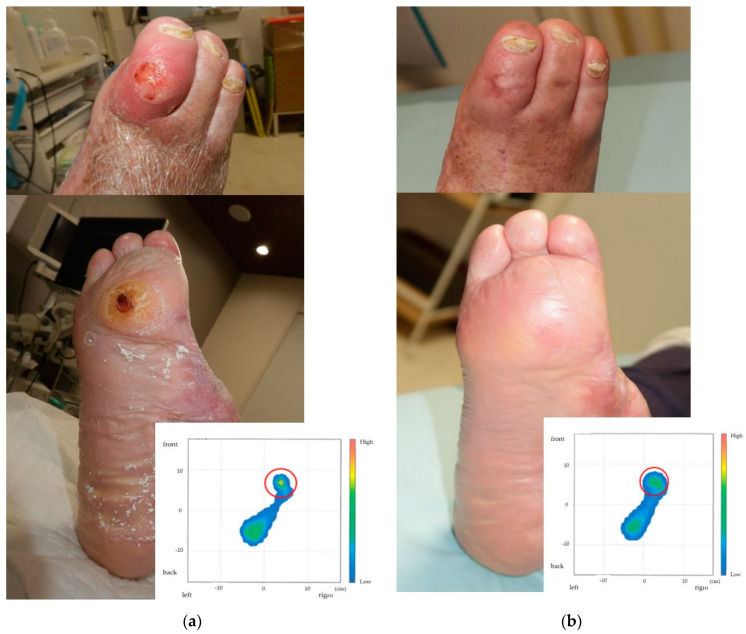
Preoperative and postoperative foot photographs and plantar pressure values. (**a**) Preoperative ulcers on the dorsal PIP joint and plantar MTP joint of the third toe, highlighting significantly elevated plantar pressure (6.61 kPa/cm^2^) (red circle). (**b**) Postoperative healing and pressure reduction to 3.54 kPa/cm^2^ (red circle).

**Figure 2 jcm-14-00646-f002:**
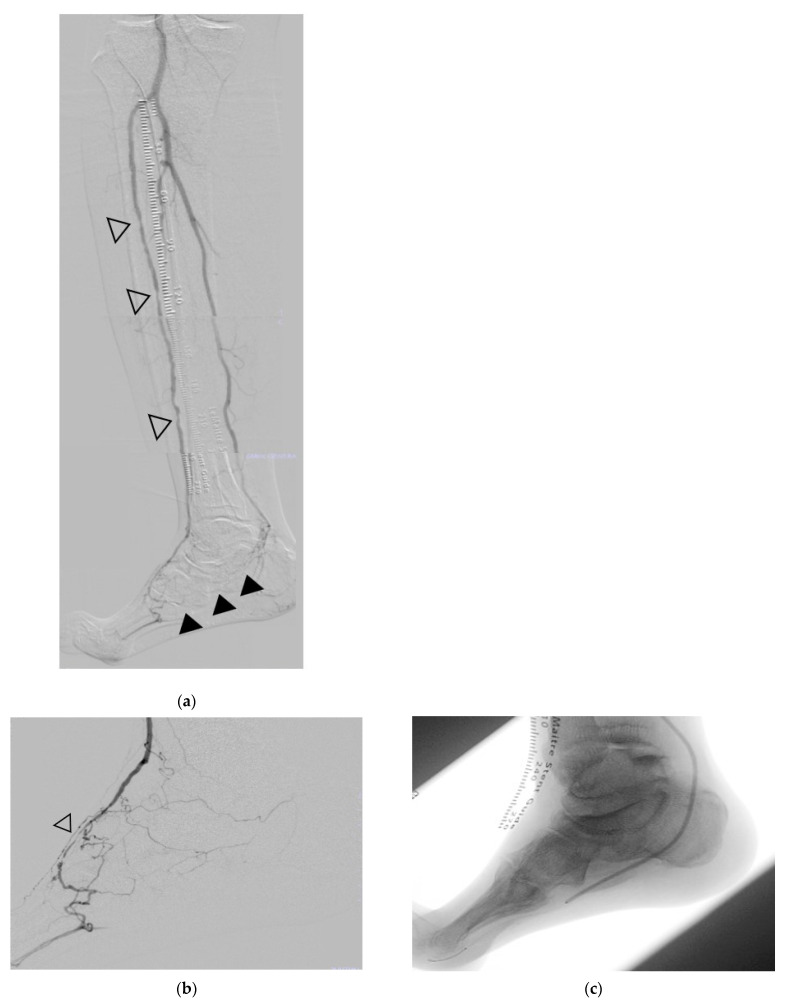
Revascularization. (**a**) Preoperative stenosis in the anterior tibial artery (outlined triangle) and dorsal foot artery (black-filled triangle). (**b**) Selective angiography of preoperative stenosis in the dorsal foot artery. (**c**) The lateral plantar artery was dilated using a 2 mm x 220 mm PTA balloon. (**d**,**e**) Postoperative angiographic findings: restored blood flow from the dorsal foot artery and the lateral plantar artery to the plantar metatarsal artery (①) and the plantar digital artery (②) of the third toe is visible.

**Figure 3 jcm-14-00646-f003:**
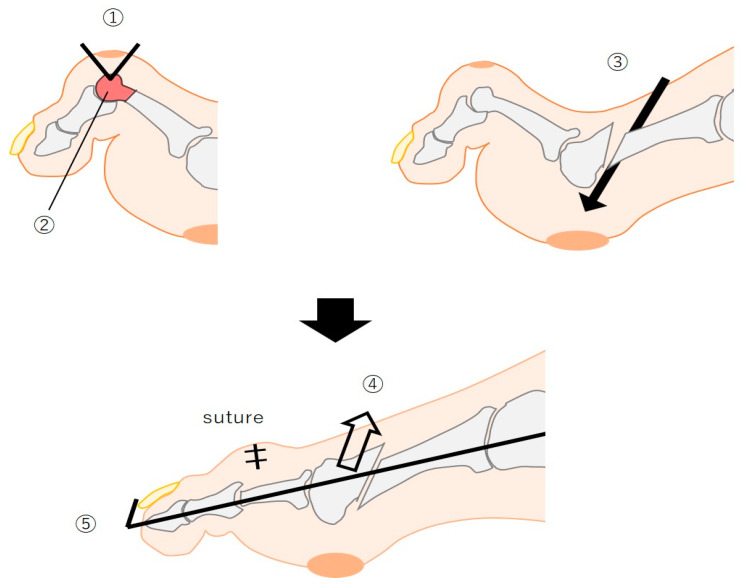
Surgical procedures. (**Upper left**) PIP resection arthroplasty showing incision (①) and resection steps (②). (**Upper right**) metatarsal shortening osteotomy. A 45°oblique osteotomy was performed at the distal third of the metatarsal (③). (**Lower figure**) the third metatarsal head was shortened by sliding it proximally (④), and internal fixation with a K-wire ensured stabilization (⑤).

## Data Availability

The data supporting the findings of this study are not publicly available due to privacy concerns. However, the data may be made available upon reasonable request to the corresponding author.

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
