# Peer review of "A Case of Diabetic Ischemic Ulcer with Toe Deformity Successfully Treated with Revascularization and Surgical Offloading"

_jcm, 2025, doi:10.3390/jcm14030646_

Round 1

Reviewer 1 Report

Comments and Suggestions for Authors

Thank you for the opportunity to review this manuscript.

The manuscript is well-written, and the clinical case is presented in a clear and structured manner. The authors have successfully described the management of a diabetic ischemic ulcer with toe deformity, showcasing the benefits of revascularization and surgical offloading in achieving excellent clinical outcomes. However, I have some comments and minor revisions to suggest before the manuscript can be accepted for publication:

  1. Innovative Contributions: While the case is well-presented and clinically relevant, it lacks significant elements of innovation. The described approach aligns with existing multidisciplinary management strategies for diabetic foot ulcers. Adding a brief discussion on how this case contributes to or enhances current knowledge, or highlighting unique aspects of the patient's management, would strengthen the manuscript's impact.

  2. Acronyms and Terminology: The manuscript uses several acronyms (e.g., EVT) without providing definitions upon their first use. For clarity and to ensure accessibility for a broader readership, I recommend including a list of acronyms with their definitions or explaining each term when it first appears in the text.

  3. Time to Healing: One critical piece of information missing is the explicit reporting of the time to healing following revascularization and surgical offloading. This information is crucial for readers to understand the clinical progression and to benchmark outcomes against other cases or studies. Please include the timeframe for ulcer healing and whether any specific factors influenced this duration.

  4. Minor Adjustments to Figures and Data: While the figures provided are informative, it would be beneficial to ensure that all visual elements (e.g., angiographic images, pressure maps) are adequately labelled and referenced in the text for seamless interpretation.

  5. In conclusion: Once the authors incorporate the suggested clarifications and additional information, it will serve as a valuable contribution to the literature on multidisciplinary management of diabetic foot complications.

Author Response

Comment 1: Innovative Contributions
While the case is well-presented and clinically relevant, it lacks significant elements of innovation. Adding a brief discussion on how this case contributes to or enhances current knowledge, or highlighting unique aspects of the patient's management, would strengthen the manuscript's impact.

Response:
Thank you for your valuable suggestion. To address this, we have expanded the discussion in the Introduction (lines 48–57) to highlight the innovative aspects of this case. Specifically, we emphasized the prioritization of in-line flow restoration and skin perfusion pressure (SPP) evaluation as key criteria for guiding surgical offloading in ischemic limbs. These parameters are underrepresented in existing guidelines and literature. Additionally, we discussed how this case demonstrates the successful integration of a multidisciplinary approach tailored to the patient’s specific clinical conditions, contributing to the understanding of individualized management strategies.

Comment 2: Acronyms and Terminology
The manuscript uses several acronyms (e.g., EVT) without providing definitions upon their first use. Including a list of acronyms or explaining each term when it first appears in the text is recommended.

Response:
We appreciate this observation. In the revised manuscript, we ensured that all acronyms, including PTA, ABI and SPP, are clearly defined upon their first use (lines 20, 23, 24, 48, 76). This adjustment aims to enhance clarity and accessibility for a broader readership.

Comment 3: Time to Healing
Explicit reporting of the time to healing following revascularization and surgical offloading is missing. This information is crucial for understanding clinical progression and benchmarking outcomes. Please include the timeframe for ulcer healing and any specific factors influencing this duration.

Response:
Thank you for this important suggestion. We have added details regarding the time to healing in the Follow-up section (lines 108–113). Specifically, we noted that the sutures were removed at two weeks post-surgery, and the K-wire was removed at three weeks post-surgery following confirmation of satisfactory wound healing and stabilization. The comprehensive postoperative management strategy, including regular dressing changes and antibiotic therapy, contributed to this timely healing process.

Comment 4: Minor Adjustments to Figures and Data
Ensure that all visual elements (e.g., angiographic images, pressure maps) are adequately labeled and referenced in the text for seamless interpretation.

Response:
We agree with your suggestion and have revised the manuscript to ensure that all figures are clearly labeled and appropriately referenced in the text (lines 81, 97, 98, 99, 103, 104, 106, 114). For example, the angiographic images (Figure 2) and plantar pressure maps (Figure 1) now include detailed captions and are explicitly mentioned in the corresponding sections of the manuscript for clarity.

Reviewer 2 Report

Comments and Suggestions for Authors

Thank you for trusting me as a reviewer of this article entitled "A Case of Diabetic Ischemic Ulcer with Toe Deformity Successfully Treated with Revascularization and 2 Surgical Offloading". 

ABSTRACT

Lines # 19-20: new ulcers at the dorsal PIP joint and plantar MTP joint of the third toe. Please write the full sentence first, followed by their abbreviations. As abbreviations are not allowed in the abstract section of the main manuscript.

Line # 22: treated with PTA. Please write the full sentence first, as previously described. And please apply this correction to any abbreviations among the whole manuscript.

Author Response

Comment 1: Abstract
Lines 19–20: "new ulcers at the dorsal PIP joint and plantar MTP joint of the third toe." Please write the full terms first, followed by their abbreviations, as abbreviations are not allowed in the abstract section of the main manuscript.

Response:
Thank you for bringing this to our attention. As suggested, we have revised the abstract (lines 19–20) to replace the abbreviations with their full terms. The updated text now reads:
"...new ulcers at the dorsal proximal interphalangeal (PIP) joint and plantar metatarsophalangeal (MTP) joint of the third toe."

Comment 2: Abbreviations in Line 22
Line 22: "treated with PTA." Please write the full term first, as previously described. And please apply this correction to any abbreviations throughout the manuscript.

Response:
We appreciate your suggestion. In line 22 and throughout the manuscript, we have ensured that all abbreviations, including PTA (percutaneous transluminal angioplasty), are introduced with their full terms upon first use. This revision ensures consistency and improves readability for a broader audience.

Reviewer 3 Report

Comments and Suggestions for Authors

Dear Authors:

Thank you for allowing me to review this manuscript. I believe that you present a good manuscript, but that it can be improved. I am enclosing some comments in the spirit of improvement. As a general comment, I recommend consulting the CAREGuidelines: Consensus-based Clinical Case Reporting Guideline Development.

Introduction: It should be improved. Only one bibliographic reference is provided. The current IWDDF recommendations on ischaemia discharge could be provided. I believe that the relevant topic of this paper is the therapeutic intervention in the presence of simultaneous ischaemia and deformity requiring offloading, and this is what should be highlighted in the introduction, supported by bibliographic references. 

2: Case report

More data should be provided on the patient's history of diabetes (type of diabetes, years since diagnosis, glycosylated haemoglobin, type of treatment; oral antidiabetics or insulin etc). It is relevant clinical information to assess a patient with diabetic foot disease.

Line 58: what meanings of acronym EVT

2.1 Erratum: must be Clinical findings, not clinical findings

2.1- It is striking that with such a large number of classifications for diabetic foot disease (Pedis, WifI, SINBAD, Texas, Wagner etc), ulcers have not been categorised in any of the existing systems. This is important because when comparing studies these systems help to evaluate the effectiveness of interventions according to category. You should therefore categorise these ulcers (not just categorise vascular status with WiFi).

2.5.-I think that the section on follow-up is poor. Data should be provided on how the post-surgical dressings were carried out. It is also necessary to indicate whether any type of plantar orthosis or special footwear was used after surgery, both in the initial post-surgical phases and in the maintenance in subsequent years; surgical offloading does not imply that subsequent offloading by means of orthopaedic systems is not necessary. Data on post-surgical perfusion levels would also help to evaluate the effectiveness of revascularisation.

Finally, I believe that the sequencing of therapeutic actions needs to be better explained. I believe that revascularisation was done first and then the unloading surgical procedure. What was the time interval between the two surgical procedures?

Author Response

  1. General Comment

Comment: Thank you for allowing me to review this manuscript. I believe that you present a good manuscript, but that it can be improved. I am enclosing some comments in the spirit of improvement. As a general comment, I recommend consulting the CARE Guidelines: Consensus-based Clinical Case Reporting Guideline Development.

Response:
Thank you for your suggestion. We have reviewed the CARE Guidelines and ensured that our manuscript aligns with their recommendations. We have included all necessary elements, such as detailed patient history, a comprehensive management plan, and explicit follow-up outcomes.

  1. Introduction

Comment: It should be improved. Only one bibliographic reference is provided. The current IWGDF recommendations on ischemia discharge could be provided. Highlight the therapeutic intervention in the presence of simultaneous ischemia and deformity requiring offloading.

Response:
We have revised the Introduction (lines 48–57) to include a discussion of the IWGDF 2023 guidelines and their recommendations on managing ischemia and mechanical stress in diabetic foot ulcers. The updated text highlights the prioritization of in-line flow restoration and SPP evaluation as key parameters to guide surgical offloading. Additional bibliographic references have been included to support these points.

  1. Case Report

Comment: More data should be provided on the patient's history of diabetes (type of diabetes, years since diagnosis, HbA1c, type of treatment, etc.). It is relevant clinical information to assess a patient with diabetic foot disease.

Response:
We have expanded the patient’s history in the Case Report section (lines 60–64). The revised text now reads:
"A 70-year-old male was diagnosed with Type 2 Diabetes Mellitus 20 years ago. At the time of presentation, the HbA1c level was 6.8%, indicating moderate glycemic control. The diabetes was managed with oral antidiabetic medications, including metformin and sulfonylurea, along with a basal-bolus insulin regimen."

  1. Abbreviation for EVT

Comment: Line 58: What does EVT mean?

Response:
To improve clarity, we replaced "EVT" with "PTA (percutaneous transluminal angioplasty)" throughout the manuscript, as PTA is more commonly used in this context (line 65).

  1. Erratum in Section Title

Comment: Line 72: Must be "Clinical Findings," not "clinical findings."

Response:
Thank you for pointing this out. We have corrected the section title to "Clinical Findings" (line 72).

  1. Ulcer Classification

Comment: It is striking that with such a large number of classifications for diabetic foot disease (e.g., PEDIS, WIfI, SINBAD, Texas, Wagner), the ulcers have not been categorized in any of these systems. This is important for comparing studies and evaluating the effectiveness of interventions.

Response:
We have categorized the ulcers using the WIfI classification, as follows:
"Based on the WIfI classification, the ulcer at the PIP joint and on the plantar aspect of the third metatarsal head can be classified as W1I2fI2 (Stage 4) and W2I2fI2 (Stage 4), respectively." This revision has been added to lines 77–79.

  1. Follow-Up Section

Comment: The follow-up section is poor. Provide data on post-surgical dressings, plantar orthosis or footwear usage, and post-surgical perfusion levels.

Response:
We have expanded the Follow-Up section (lines 108–124) as follows:
"After surgery, a standardized dressing protocol was implemented, using non-adherent hydrocolloid dressings changed every 3 to 4 days under sterile conditions. Antibiotic therapy was administered for 10 days to prevent infection. Sutures were removed at two weeks post-surgery, and the K-wire was removed at three weeks post-surgery. Therapeutic sandals were used during the initial postoperative phase to offload pressure from the surgical sites. Once complete healing was achieved, custom-made orthopedic shoes with pressure-redistributing insoles were provided for long-term maintenance. ABI remained within normal limits throughout the three-year follow-up, confirming sustained perfusion."

  1. Sequencing of Therapeutic Actions

Comment: The sequencing of therapeutic actions needs to be better explained. I believe that revascularization was done first and then the unloading surgical procedure. What was the time interval between the two surgical procedures?

Response:
We have clarified the sequencing of therapeutic actions in the Management Plan section (lines 84–94):
"Revascularization was prioritized to improve blood flow to the ulcerated area, as insufficient perfusion was identified as the primary barrier to wound healing. Post-procedure, the patient’s SPP values normalized within 48 hours, confirming successful revascularization. Surgical offloading was performed one week later, after ensuring infection control and adequate perfusion. These decisions were made following a multidisciplinary discussion to ensure optimal outcomes."